# 3D PEEK Objects Fabricated by Fused Filament Fabrication (FFF)

**DOI:** 10.3390/ma15030898

**Published:** 2022-01-25

**Authors:** Inwoo Baek, Oeun Kwon, Chul-Min Lim, Kyoung Youl Park, Chang-Jun Bae

**Affiliations:** 1Department for 3D Printing Materials, Korea Institute of Materials Science, Changwon 51508, Korea; iwbaek@kims.re.kr (I.B.); bapkoe0412@gnu.ac.kr (O.K.); 2Defense Space Technology Center, Agency for Defense Development, Daejeon 34186, Korea; cmlim@add.re.kr (C.-M.L.); kypark@add.re.kr (K.Y.P.)

**Keywords:** 3D printing, fused filament fabrication, poly ether ether ketone, LEO aerospace, Weibull modulus, reliability

## Abstract

PEEK (poly ether ether ketone) materials printed using FFF 3D printing have been actively studied on applying electronic devices in satellites owing to their excellent light weight and thermal resistance. However, the PEEK FFF process generated cavities inside due to large shrinkage has degraded both mechanical integrity and printing reliability. Here, we have investigated the correlations between nozzle temperatures and PEEK printing behaviors such as the reliability of printed line width and surface roughness. As the temperature increased from 360 to 380 °C, the width of the printed line showed a tendency to decrease. However, the width of PEEK printed lines re-increased from 350 to 426 μm at the nozzle temperatures between 380 and 400 °C, associated with solid to liquid-like phase transition and printed out distorted and disconnected lines. The surface roughness of PEEK objects increased from 49 to 55 μm as the nozzle temperature increased from 380 to 400 °C, where PEEK is melted down and quickly solidified based on more energy and additional heating time at higher printing temperatures at 400 °C. Based on these printing trends, a reliability analysis of the printed line was performed. The printed line formed the most uniform width at 380 °C and had a highest Weibull coefficient of 28.6 using the reliability analysis technique called Weibull modulus.

## 1. Introduction

PEEK, one of the most important replacement materials for metal in high performance, is produced by step-growth polymerization reaction as a semi-crystalline resin that has high mechanical strength, chemical resistance, and thermal stability [1], especially thermal properties where the glass transition (T_g_) and melting temperatures (T_m_) are 143 and 343 °C, respectively, and these temperatures are satisfied with the thermal stability required for the end-use application such as medical implants, energy storage, military engineering and aerospace, etc. For example, low earth orbit (LEO) satellites are being actively developed in the new space age by not only national research labs but also privately-led space industries such as Blue Origin and SpaceX [2]. However, since the traditional metal components are too heavy and are a big bottleneck on manufacturing cost as well as operational lifetime, researchers are looking for a way to make the components light, cheap, and long-duration from the commercial standpoint [3]. The research actively being performed is super engineering plastic, which has distinct properties such as high strength, excellent fracture toughness, and heat resistance compared to conventional plastics [4,5,6].

Given the superior properties of PEEK, additive manufacturing (AM) is fabricating PEEK-based three-dimensional objects to apply various applications. AM is the technology that uses computer-aided design (CAD) systems with user convenience by simulation capabilities, and it also has the potential for price competitiveness by cost reduction and time saving due to increasing fabrication speed [7]. The process is divided into seven categories according to materials and processes in the ISO/ASTM 52900:2021, which are photo-polymerization, material jetting, binder jetting, material extrusion, powder bed fusion, sheet lamination, and directed energy deposition [8]. Among categories, selective laser sintering (SLS) as powder bed fusion process is widely used in the heavy industries such as automotive design, aerospace parts, and medical devices. As the popular reason of commercial sector, SLS printing is attractive because various powder materials are able to print, and the post processes are simpler than other printing methods. However, PEEK materials have experienced distortion or deformation when high power is exposed to sinter a thermoplastic resin powder of PEEK in SLS printing due to a higher shrinkage rate, 2.0%, than other plastics [9]. Kim et al. reported that PEEK printed using the SLS technique includes a structural cavity that affects the roughness and stiffness. Modeling of PEEK cavity showing that when the temperature was decreased by 10 times, the intermolecular attraction prevailed rather than the heat disturbance of PEEK molecules, and the overall volume of the structure decreased [10]. Another category, the SLA method, also has been reported to fabricate a PEEK structure, which was manufactured using PEEK powder and photosensitive resin in the form of a green body. PEEK powder contained 37% in green body and, when sintered 275 °C for 1 h, the Young’s modulus and hardness are 808 MPa and 59.0 HRF, respectively. However, stereolithography required the binder-burn out process and sintering, and there is a limitation to manufacturing commercial and bigger parts as well as mechanical parts [11].

Compared to other techniques, the fused filament fabrication (FFF) process has an advantage in the economy due to only heating, melting, and additive manufacturing the filaments. This method is applied to manufacture medical devices and aerospace parts, and in particular, in the medical field, PEEK material is being used in the manufacture of implants and medical devices. PEEK-15wt% cHAp biocomposite was used for a medical bone implant, showing 37% better cell activity than traditional [12]. FFF printing melts and extrudes PEEK filaments, starting crystallization and shrinkage, which affect the surface roughness and strength, causing cracks and fatigue failure. In order to prevent warpage and improve the precision and bonding strength of PEEK, Bin et al. reported that heat treatment is used to minimize shrinkage from 20.4% to 5.0% of PEEK parts edge as well as to strongly attach to the floor [13]. In another case, a nanosat satellite designed by Rinaldi et al. is manufactured using the PEEK FFF 3D Printing method with considerations of thermal analysis and outgassing performance, 0.19% as TML, which is a satisfied NASA standard [14,15]. However, there is a lack of fundamental research on how to interconnect printing parameters to PEEK printing behavior. These above cases were only analyzed object characteristics such as surface roughness and mechanical strength, from when manufacturing was completed, but with an absence of material reliability.

Herein, we have investigated the correlations between nozzle temperatures and PEEK printing behaviors such as printed line width and surface roughness and analyzed the reliability of these phenomena. Given the nozzle temperatures ranging from 360 to 400 °C, the line widths and surface roughness of 3D-printed PEEK were measured and the uniformity was evaluated through quantitative analysis. These results show that control of the thermal process condition has tremendous potential to avoid cracks, increase the quality of parts, and avoid reliability issues. 

## 2. Materials and Methods

### 2.1. FFF Method to Print PEEK

In this research, FFF 3d printer (Shark-Pro, K-Labs, Ulsan, Korea) nozzle temperature can be increased up to 400 °C, and the chamber is set at 110 °C during printing, as shown in Figure 1a. The PEEK filament (RB PEEK, Hanil Industrial Company, Namyangju, Korea) melting temperature is 334 °C, and the glass transition temperature (T_g_) is 143 °C and dried in a 100 °C oven for 5 h to prevent the absorption of water. Figure 1b shows that dried PEEK is extruded through the heating nozzle and laminated layer-by-layer above the metal plate with a lamp, which is a source of heat. Samples were prepared by different nozzle temperatures, which is the most influential factor for the final output.

As shown in Figure 1c, we fabricated a structure with different nozzle temperatures (T_N_) set to 360, 370, 380, and 400 °C, and the chamber temperature (T_C_) was 100 °C, printing speed (v) was 15 mm/s, and layer thickness (t) was set to 200 μm. The sample was sliced by the Ultimaker Cura software program after making an STL file to 200 × 200 × 100 mm^3^, an infill percentage of 60%, and a wall thickness of 400 μm. In order to check the effect of each factor on the width of the contact area of the printed lines, the monolayer between 290th and 300th layer was detached and the width of the contact area was measured. Figure 1d is in the form of an outer wall and an inner wall attached to each other, and in order to improve adhesion when filling the inside, the influence of the overlapping part of the printed lines in the red area was measured through an optical microscope.

### 2.2. Surface Roughness Measurement

The surface roughness of PEEK parts was measured according to ISO standard 21920-2:2021 [16] using a stylus profilometer (Dektak XT, Billerica, MA, USA). The radius of the stylus was 25 μm and the movement speed was 1 mm/s, which was determined by the ISO standard.

Surface roughness on the side section of FFF samples, printed at Z-axis, is investigated with two different methods: line-drawing and 3-D mapping, respectively. In the line-drawing method, a stylus profiler repeated three different points to measure the side parts of samples. We used a maximum surface roughness ($R_{z})$ of a ten points height of irregularities that cut-off after the 2.5 mm filter. This filter can be separate the graph into three parts: nominal form, waviness, and roughness. After deleting waviness in a nominal form, then roughness can be measured.

Sampling length is determined by roughness profile, which, in this case, is 12.5 mm due to the roughness range including 10 to 50 μm.
(1)$$R_{z}=\frac{\left(h_{1}+h_{2}+h_{3}+h_{4}+h_{5}\right)-\left(v_{1}+v_{2}+v_{3}+v_{4}+v_{5}\right)}{5}$$
where $R_{z}$ is maximum surface roughness, *h* is the highest hill point, and *v* is the lowest valley point of the roughness profile measured in equipment.

In 2-D and 3-D mapping analysis, a stylus profiler was repeated 250 times during the increase of 10 μm to measure the total area of the side part of the Z-axis sample plane. After profiler moving, the nominal form also can be filtered to remove waviness by a cut-off of 2.5 mm after measuring and re-drawing the 3-D mapping. An image of the total area describes surface roughness between the hill and valley using different colors; the highest area is represented in red and the lowest is in blue.

## 3. Results and Discussion

### 3.1. Dimensional Deformation of 3D Printed PEEK Filaments

PEEK has experienced a larger shrinkage rate than other super engineering plastics such as polyphenylene sulfide and polyimide [9,17]. The dimension of FFF objects is directly dependent on the printing conditions such as nozzle temperature (T_N_), chamber temperature (T_C_), printing speed (v), and layer thickness (t), respectively. For example, when PEEK filaments are printed at the higher printing temperature [18,19], the filaments activated by large thermal energy become more viscous and wide, which increases the instability of fabricating PEEK-based 3D objects. Therefore, in order to precisely control the shrinkage, defining the nozzle condition is very important because the temperature parameter directly affects not only mechanical properties such as surface roughness and tensile strength but also printing quality and reliability of the total product.

Figure 2 shows the line widths of PEEK filaments fabricated using FFF printing at the different nozzle temperatures. Note that the chamber temperature is fixed around 100 °C, and all printed samples are cooled down at atmospheric conditions. As shown in Figure 2a,b, the widths of the printed lines contacted in consequential layers measured through an optical microscope are plotted as a function of nozzle temperatures, respectively. The result shows that printed line width of 550 μm is observed at the nozzle temperature of 360 °C when original PEEK filaments of 1.7 mm are printed out of the FFF nozzle of 400 μm. In addition, widths of PEEK lines are inversely proportional to the nozzle temperatures between 360 and 380 °C, decreased from 550 to 350 μm, which is associated with the shape deformation above the PEEK melting temperature of 345 °C. This phenomenon can be explained by the competitive relationship between shrinkage and diffusion. In this case, the shrinkage rate of the extruded PEEK at a high nozzle temperature is larger than the diffusion time to bond between printed lines. According to Wang et al., surface topography can show voids among filaments. PEEK shrinkage is a more dominant factor than diffusion time of printed lines since the bond neck cannot be formed at a temperature below 230 °C [20]. Given the further increase of nozzle temperature up to 400 °C, the filaments exposed to a severe environment have experienced a phase transition from solid to liquid-like and printed out distorted and disconnected lines, as shown in Figure 2b. In additional, the diffusion time to bond between printed lines was sufficiently given to increase the width due to more kinetic energy of the movement.

Table 1 shows the average width and deviation of PEEK lines in the sequential layers printed at the different nozzle temperatures ranging from 360 to 400 °C. The filaments printed at the nozzle temperature of 360 °C are not melted, thus experiencing stress accumulation when they are extruded through nozzle size of 400 μm and developing the largest width of 556 μm and deviation of 36 μm. On the other hand, the filaments melted at the temperature of 380 °C has been relaxed in the stress, then printing the smallest width of 351 μm and deviation of 15 μm.

### 3.2. Surface Roughness of 3D Printed PEEK Objects

The surface roughness of PEEK objects was investigated as a function of nozzle temperature, one of the main parameters of the FFF process, which is associated with melting thermoplastic polymer and laminating sequential PEEK layers. Figure 3a,b show a side-sectional image of PEEK objects built at the Z-axis stacking direction at the red box of Figure 1c and a line-drawing profile measured using a stylus profilometer.

Note that there are interesting observations where the surface roughness of PEEK objects decreased from 60 to 54 μm, and then the surface roughness increased up to 67 μm as the nozzle temperature increased from 360 to 400 °C in Figure 3c. The thermal behavior can be explained by Wang et al., the higher temperature can improve fluidity cause the density to increase from 88% to 90% to fill voids inside the part [21]. Given more energy and additional heating time at higher printing temperatures at 400 °C, PEEK is melted down and quickly solidified passed through printing nozzle, resulting in larger rough surfaces of 55 μm compared to the printing temperature of 380 °C.

The 3D mapping technique in Figure 3d is applied to analyze the surface roughness of cross-sectional images of PEEK objects, measuring the roughness on the distance between hill and next following hill in sequential stacking layers in which the roughness was averaged with 1000 times measurements. As shown in Figure 3d, 3D mapping targets consist of hills (red color) and valleys (blue color), and the surface roughness equivalent to the difference between hills and valleys is approximately 55 μm.

### 3.3. Reliability of 3D Printed PEEK Lines

In FFF printing, one of the important factors is how to control printed line width, with which the final quality of the product is directly associated. For example, if the line width of each layer can be uniformly maintained, the FFF process not only easily prints the line closer to the nozzle size but also accurately reproduces high-quality 3D complex structures. On the other hand, non-uniform printed lines cause inhomogeneous contact areas in the stacking layers and defects such as pores and cracks, which are fatal to mechanical properties. According to Ding et al., for impact strength of printed PEEK with different nozzle temperatures in the range of 360 to 420 °C, the highest value is 67 kJ/m^2^, printed at 380 °C [22]. Moreover, the surface roughness, the main factor of final quality, can be affected by non-uniform printed lines. Ensuring the way to meet the requirement of commercial manufacturing is related to developing uniformly controlled printed lines and preserving the lower deviation in the width of the contact area as much as possible.

Figure 4 presents the reliability of 3D printed PEEK lines investigated with the line width of extruded filaments and shape deformation as a function of different nozzle temperatures. As shown in Figure 4a, the widths and shape deformation of extruded lines are averaged from 12 samples and calculated from the ratio of width to nozzle diameter of 400 μm. The filament extruded through the FFF nozzle has an irregular width from 630 to 500 μm due to not completely melting at the relatively low temperature of 360 °C. On the other side, as the temperature increased up to 370 °C, the widths and shape deformation of extruded filaments, 400 μm and 1.0, are closer to the nozzle diameter of 400 μm. Uniform width and best uniformity achieved at 370 °C are related to the gradual transition from elastic to viscous behavior. After that, it was observed that the overall uniformity was decreased due to line breakage at 400 °C, and the line width was increased to 426 μm.

Figure 4b shows that the Weibull analysis is performed to measure the width of printed lines as a function of different nozzle temperatures. The Weibull coefficient value is calculated to determine what temperature is an optimum printing condition to print uniform lines. As the nozzle temperature increased from 360 to 370 °C, the Weibull coefficient highly increased from 17.2 to 28.6 and then gradually decreased to 19.2 at 400 °C. At the temperature of 360 °C, the PEEK filament was not completely melted through the nozzle, experiencing stress accumulation, so there are the largest deviations of the printed lines as well as the lowest Weibull coefficient of 17.2, which represents a non-reliable printing condition. At the temperature of 370 °C, however, the filaments have sufficiently been melted and uniformly extruded, and they have a small deviation of printed line width as well as a large Weibull coefficient of 28.6, which is the most reliable printing condition and almost two times more reliable than the temperature of 360 °C. After that, the reliability gradually decreases due to line breakage and drops to 19.2 at 400 °C.

## 4. Conclusions

In this study, the co-relationship between nozzle temperature and printing behavior such as line width and surface roughness, which affected reliability, was investigated. The filaments activated by large thermal energy become more viscous and wide, which increases the instability of fabricating PEEK-based 3D objects when PEEK filaments are printed at the higher printing temperature. In order to precisely control the expansion of shrinkage of PEEK, it is very important to find the correlations between nozzle temperatures and PEEK printing behaviors such as reliability of printed line width and surface roughness. The potential for controlling the expansion of shrinkage rate was experimentally shown according to the nozzle temperature, which is affecting the line width and surface roughness. Line widths of extruded PEEK are inversely proportional to the nozzle temperatures between 360 and 380 °C, decreased from 550 to 350 μm, which is associated with the shape deformation above the PEEK melting temperature of 345 °C. This phenomenon can be explained by the fact that the greater shrinkage rate of the extruded PEEK on high nozzle temperature is bigger than the diffusion time to bond between printed lines. Given the further increase of nozzle temperature up to 400 °C, the filaments exposed to a severe environment have experienced a phase transition from solid to liquid-like and printed out distorted and disconnected lines. The surface roughness of PEEK objects decreased from 56 to 49 μm, and then the surface roughness increased up to 55 μm as the nozzle temperature increased from 360 to 400 °C, respectively. Given more energy and additional heating time at higher printing temperatures at 400 °C, PEEK is melted down and quickly solidified when passed through the printing nozzle, resulting in larger rough surfaces of 55 μm compared to the printing temperature of 380 °C.

Furthermore, the reliability of the uniformity line widths was analyzed through the Weibull modulus. The Weibull coefficient had the highest value of 28.6 at 380 °C, indicating that the line was extruded most uniformly under this nozzle temperature. To evaluate the relationship between nozzle temperature and mechanical characteristics, a study on the tensile strength will be carried out in the future.

## Figures and Tables

**Figure 1 materials-15-00898-f001:**
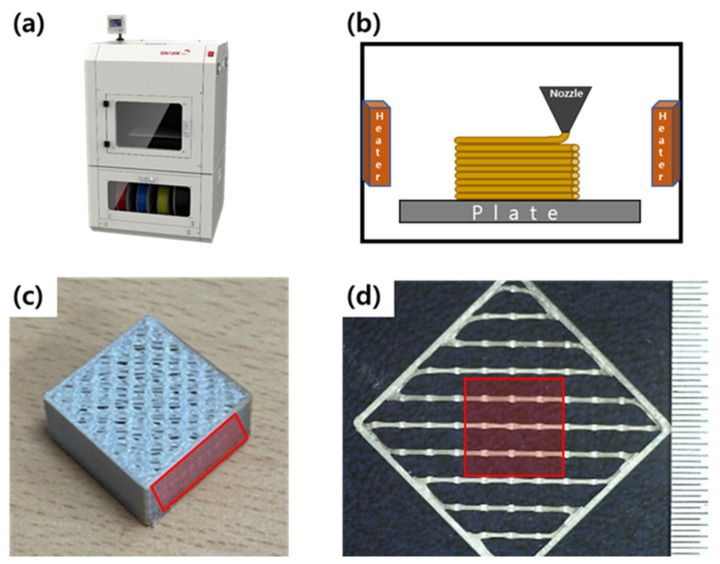
Specimens for measurement line width: (**a**) fused filament fabrication (FFF) system, (**b**) scheme of 3D printing process, (**c**) 3D Printed PEEK, and (**d**) mono-layer between PEEK printed lines.

**Figure 2 materials-15-00898-f002:**
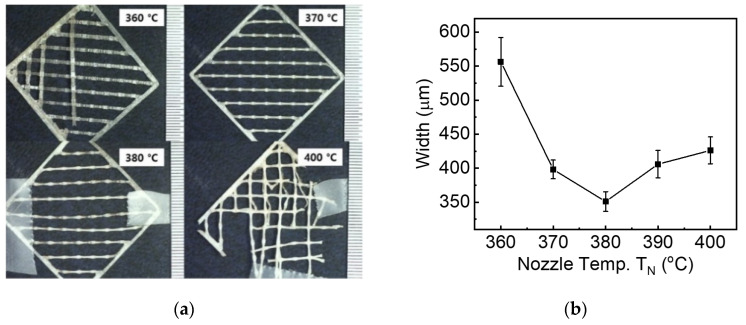
Width of PEEK lines printed at the different nozzle temperatures (T_N_). (**a**) Optical images showing the width of the printed lines contacted in consequential layers; (**b**) Line width as a function of nozzle temperature printed at the fixed chamber temperature of 100 °C.

**Figure 3 materials-15-00898-f003:**
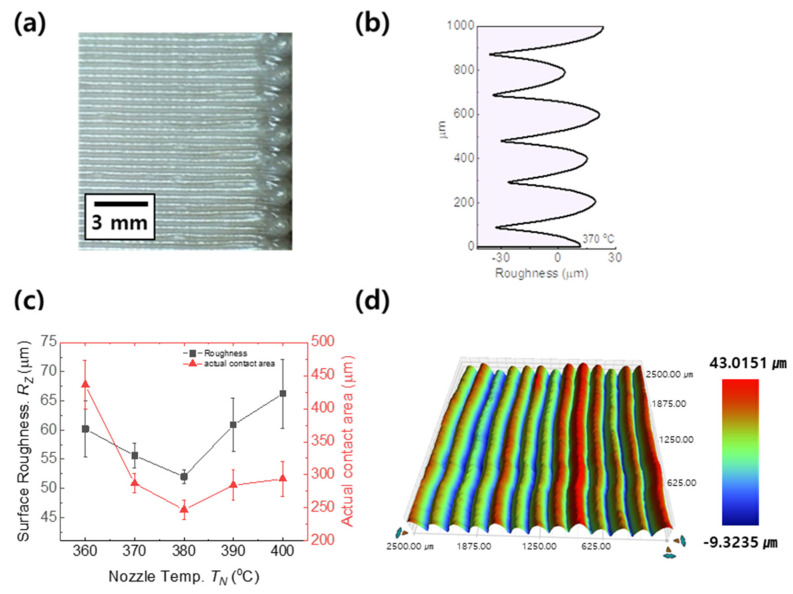
The surface morphology of side parts on specimens: (**a**) side view of the printed structure, (**b**) 2-D surface roughness at 370 °C, (**c**) comparison on surface roughness between different T_N_, and (**d**) 3-D mapping image of surface morphologies.

**Figure 4 materials-15-00898-f004:**
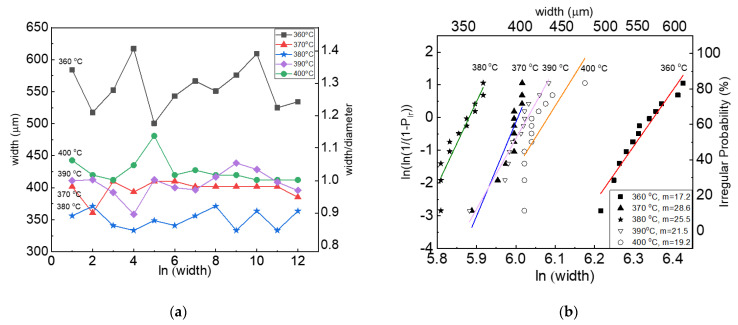
Analysis of printed line reliability used by Weibull modulus: (**a**) Measurement value ratio of line width and nozzle diameter, (**b**) Weibull plot of the line width with various nozzle temperatures.

**Table 1 materials-15-00898-t001:** Width and deviation of nozzle temperature.

T_N_	Width (μm)	Deviation
360 °C	556.47	35.73
370 °C	398.23	13.75
380 °C	351.04	14.56
390 °C	406.01	20.14
400 °C	426.22	19.76

## Data Availability

All data have been included in the paper.

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
