# Peer review of "3D PEEK Objects Fabricated by Fused Filament Fabrication (FFF)"

_materials, 2022, doi:10.3390/ma15030898_

Round 1
Reviewer 1 Report
The authors present experimental results examining the influence of nozzle temperature on the quality of 3D printed PEEK. While not particularly glamorous, work like this is vital to improving manufacturing capabilities, and I appreciate the effort put in by the authors. Prior to publication, I recommend considering the following:
- It should be noted that Fused Deposition Modeling (FDM) is trademarked to Stratasys. It might be best to use the more generic term of Fused Filament Fabrication (FFF).
- There are several parts where the language is somewhat awkward. The first sentence of the abstract is an example where the statement is decipherable but confusing upon first read. Aside from the strange sentence structure, the first sentence mentions "possessed of cavities." Are you referring to the cavities caused during SLS printing as mentioned later in the document?
- Capitalize "Origin" in line 24.
- Define Tg in line 70. In the same line, at what temperature was the filament dried?
- It is somewhat unclear where the measurements are taken. Perhaps include a diagram or mark an image showing where exactly roughness is measured and where line width is measured.
- A "mono layer" was detached and used to measure line width. How was this mono layer removed? Where is width measured? I assume it was a layer in the middle of the sample rather than the top or bottom layer.
- Line 150-151 describes figure 3a and 3b as cross-sectional. Is it actually a cross-sectional image, or is it just a side view? Where, physically on the part, is the roughness measured?
- Line 80 mentions an STL file. Specify that it is a simple box. Maybe also mention the infill percentage and wall thickness.
- In a few instances (such as line 120 and caption of Figure 1d), a 3D printed line is referred to as a "filament." This is confusing language since the unprocessed 3D print material is generally called filament. Perhaps refer to them as "printed line" or "layer" or similar.
- I'm a little confused as to what figure 3d is showing. I thought it was just the metrology of the printed part (the side). Why is it flattened (like a plateau) rather than rounded like figure 3b?
- It would be good to show images of 3D printed parts using each temperature. Given that the motivation of this work is to improve print quality and reliability, a mechanical testing of these parts would go even further to strengthen the paper, though this not strictly necessary.
Author Response
Thank you for your kind comments.
Please see the attachment.

Reviewer 2 Report
The article "3D PEEK objects fabricated by Fused Deposition Modeling 2 (FDM)" is interesting.
Few suggestions are given below to improve the quality of the article.
- Abstract is not perfect. Importance of the study and major findings are not presented.
- The introduction part is too short containing limited information regarding importance and need of this study.
- Results are presented without scientific reasons. Also, the comparison with the exsiting studies is not presented.
- Conclusions are almost similar to the abstract. Its suggested to draw conclusios in better way.
Author Response

(The authors gave the same response as above.)

Reviewer 3 Report
There are some language issues (i.e., line: 42: "and elastic and urethane parts"; line 133 - used form can't)
- Line: 23: If you use abbreviations, first you need to attach the full name - low earth orbit and then the proper abbreviation (LEO).
- Line 44: In the SLS technology, laser power is equal to about 40W at peak value so it is not high power.
- Line 49: hot nozzle is not dedicated to shrinkage reduction in the FDM technology. There are completely different solutions for the reduction of that issue.
- In your citations, there are not any quantified values in the presented results. Additionally, in your introduction, there is a novelty indication missing related to your research.
- The introduction is too general and too short. Also, the literature review needs to be significantly improved.
- Line 70: what is Tg? Please put data about drying.
- What was the reason for using such temperature intervals? Why do authors resign from the analysis 390℃ setting? From your research point of view, it is a very important point, because there is a significant curve course change in the chart - figure 2b
- The authors sentenced about shrinkage based on some literature review - it should be moved into the introduction part. In the results and discussion chapter, you should focus on your original research.
- Please put more data on how to control the
shrinkage by defining nozzle condition. - There is no information why the authors chose only one outline shell and such a small infill. If you were analyzing surface roughness you should make the external surface more stable.
To summarize, the manuscript has no correct structure, and its scientific value is too low. There should be more additional research related to material analysis. That is why I suggest rejecting this manuscript.
Author Response

(The authors gave the same response as above.)

Round 2
Reviewer 3 Report
Dear Authors,
You made significant improvements, but in my opinion, you should take into account the following comments:
- You put the PEEK abbreviation without an explanation of what this abbreviation means.
- You analyzed the shrinkage after printing only one layer, please explain why? The first layer is connected with different material (the substrate plate material) and it behaves significantly different as it is in the connection between the next PEEK layers.
- Based on figure 3 analysis it could be concluded that authors made only one hardness measurement without any average values calculations. It should be some statistical analysis included.
- In my opinion, the scientific value of this manuscript is quite low, maybe the authors could attach some microscopic investigation?
Author Response
Response to Reviewer 3 Comments
Point 1: You put the PEEK abbreviation without an explanation of what this abbreviation means.
Response 1: PEEK is abbreviation of “poly ether ether ketone” which is step-growth polymerization polymer as explained in line 9.
Point 2: You analyzed the shrinkage after printing only one layer, please explain why? The first layer is connected with different material (the substrate plate material) and it behaves significantly different as it is in the connection between the next PEEK layers.
Response 2: Thanks for clarifying the important point. We had also the same consideration on the different materials between the first PEEK layer and metal substrate which reviewer pointed out so that we actually investigated to the line widths between 290 and 300th layers as a function of printing temperature.
Point 3: Based on figure 3 analysis it could be concluded that authors made only one hardness measurement without any average values calculations. It should be some statistical analysis included.
Response 3: As your pointed out, we performed the experiment and plotted the result of deviation in figure.3(c) as explained in line 184.
Point 4: In my opinion, the scientific value of this manuscript is quite low, maybe the authors could attach some microscopic investigation?
Response 4: Our draft’s novelty and scientific value are not to only investigate the correlations between nozzle temperatures and PEEK printing behaviors such as printed line width, surface roughness, but also analyze the reliability of these phenomenon. Compared to pre-existing references, our new observation will provide a guild-line of the process when 3D PEEK objects are fabricated by Fused Filament Fabrication (FFF).
